# Human Peripheral Blood Dendritic Cell and T Cell Activation by *Codium fragile* Polysaccharide

**DOI:** 10.3390/md18110535

**Published:** 2020-10-27

**Authors:** Wei Zhang, Juyoung Hwang, Hae-Bin Park, Seong-Min Lim, Seulgi Go, Jihoe Kim, Inho Choi, Sangguan You, Jun-O Jin

**Affiliations:** 1Shanghai Public Health Clinical Center, Shanghai Medical College, Fudan University, Shanghai 201508, China; weiwei061215@126.com (W.Z.); jyhwang5@yu.ac.kr (J.H.); phb02104@yu.ac.kr (H.-B.P.); 2Department of Medical Biotechnology, Yeungnam University, Gyeongsan 38541, Korea; i0473543@ynu.ac.kr (S.-M.L.); 21421256@ynu.ac.kr (S.G.); kimjihoe@ynu.ac.kr (J.K.); inhochoi@ynu.ac.kr (I.C.); 3Research Institute of Cell Culture, Yeungnam University, Gyeongsan 38541, Korea; 4Department of Marine Food Science and Technology, Gangneung-Wonju National University, 120 Gangneung Daehangno, Gangneung, Gangwon 210-702, Korea

**Keywords:** *Codium fragile* polysaccharide, BDCA1 PBDC, BDCA3 PBDC, helper T cell, cytotoxic T lymphocyte

## Abstract

Natural polysaccharides exhibit an immunostimulatory effect with low toxicity in humans and animals. It has shown that polysaccharide extracted from *Codium fragile* (CFP) induces anti-cancer immunity by dendritic cell (DC) activation, while the effect of CFP has not examined in the human immune cells. In this study, we found that CFP promoted the upregulation of CD80, CD83 and CD86 and major histocompatibility complex (MHC) class I and II in human monocyte-derived dendritic cells (MDDCs). In addition, CFP induced the production of proinflammatory cytokines in MDDCs. Moreover, CFP directly induced the activation of Blood Dendritic Cell Antigen (BDCA)1^+^ and BDCA3^+^ subsets of human peripheral blood DCs (PBDCs). The CFP-stimulated BDCA1^+^ PBDCs further promoted activation and proliferation of syngeneic CD4 T cells. The CFP-activated BDCA3^+^ PBDCs activated syngeneic CD8 T cells, which produced cytotoxic mediators, namely, cytotoxic T lymphocytes. These results suggest that CFP may be a candidate molecule for enhancing immune activation in humans.

## 1. Introduction

Marine natural products are promising sources of biologically active molecules [1,2]. Various biological modulating compounds have been extracted from marine organisms, such as sponges, algae, and mollusks [3,4]. In the field of immune modulation, there are a number of well-studied algae-derived polysaccharides, such as fucoidan, ascophyllan, carrageenan, and porphyran [5,6,7,8,9,10]. In previous studies it was found that fucoidan (extracted from brown algae) promoted the activation of dendritic cells (DCs), T cells and natural killer (NK) cells, which further induced anti-cancer immunity [10]. In addition, fucoidan has been found to elicit the activation of human peripheral blood DCs (PBDCs) [11]. Polysaccharides extracted from brown algae have been well-studied with respect to immune activation in humans and mice, whereas green algae-extracted polysaccharides have not. Moreover, the efficacy of brown algae-derived polysaccharides has not been compared with that of green algae-extracted polysaccharides.

*Codium fragile*, a green alga, is a traditional Asian food ingredient [12]. Sulfated polysaccharides purified from *C. fragile* exhibit numerous biological effects, including anti-obesity, anti-viral, and anti-cancer effects [13,14,15]. *C. fragile* polysaccharide (CFP) has also shown immunostimulatory effects. In a previous study, it was shown that fraction 2 of the sulfated polysaccharide has a stronger effect in the induction of NK cell activation in vitro [16]. Moreover, CFP promoted splenic DC activation in vivo in a mouse model, and this activation in turn elicited antigen-specific immune activation and anti-cancer effects [5]. These CFP-activated immunostimulatory effects have yet to be reported in human immune cells. In addition, the DC-activating effect of CFP (in humans) has not been compared with that of fucoidan. 

DCs are the most powerful antigen-presenting cells that induce T cell immunity [17,18]. Activation of DCs is defined by the upregulation of co-stimulatory molecules, presentation of antigens by major histocompatibility complex (MHC) proteins, and production of cytokines [5]. These activated DCs are called matured DCs. In humans, the stimulatory effect of a novel reagent is normally examined in monocyte-derived DCs (MDDCs), because it is easy to obtain enough cells and to differentiate the cells from monocytes. However, MDDCs and in vivo DCs are different in terms of their morphology and function [19]. MDDCs represent a single cell type, whereas human peripheral blood DCs (PBDCs) contain two main subsets, namely conventional DCs (cDCs) and plasmacytoid DCs (pDCs) [20]. The cDCs are further divided into Blood Dendritic Cell Antigen 1^+^ (BDCA1^+)^ and BDCA3^+^ cDCs. These BDCA1^+^ and BDCA3^+^ cDCs have different functions in the presentation of antigens to T cells [20,21]. The BDCA1^+^ cDCs present exogenous antigen on MHC class II, which is recognized by CD4 T cells, whereas BDCA3^+^ cDCs have the special capacity to present cytosolic and endogenous antigen on MHC class I, which induces activation and differentiation of CD8 T cells [22]. In contrast with PBDCs, MDDCs cannot be divided into subsets; consequently, it is not possible to evaluate MDDC subset-dependent T cell activity. 

Previous to our study, it was shown that CFP induced DC activation in vivo in a mouse model [23]. However, the effect of CFP in human DCs, especially on human PBDC activation, has not been studied. In addition, the effect of immune activation by CFP has not been compared with fucoidan, which is the most thoroughly investigated marine polysaccharide in terms of DC activation. We carried out the present study to test the hypothesis that CFP can induce human DC activation, which elicits T cell immunity.

## 2. Results

### 2.1. Monosaccharide Composition of CFP

Fucoidan is known to contain several types of monosaccharides. Because the goal was to evaluate immunostimulatory effects in human DCs, we first compared the monosaccharide composition of CFP with that of fucoidan. Both CFP and fucoidan contain sulfate groups (Figure 1). Fucoidan from *Fucus vesiculosus* contained high rates of fucose as the main component (Table 1). In addition, galactose, xylose, mannose, and glucose were present in fucoidan as minor components (Table 1). In CFP, Man was the main component and glucose was also present as a minor component; however, fucose, xylose, and galactose were not found (Table 1).

### 2.2. CFP Induces Activation of Human MDDCs

In a previous study, it was found that CFP induced DCs activation in a mouse model; in the present study we examined the effect of CFP in human MDDCs. Before evaluation of the DC stimulatory effect of CFP, we examined cytotoxic effect of CFP in RAW264.7 cells and HL-60 cells. The treatment of CFP did not promote cytotoxicity in RAW264.7 and HL-60 cells (Appendix A). Next, we examined the effect of CFP in the MDDCs and the MDDCs were generated from isolated human blood monocytes by culturing with recombinant human granulocyte-macrophage colony-stimulating factor (rhGM-CSF) and recombinant human interleukin (rhIL)-4 for 6 days. The MDDCs were further treated with 10, 25, 50, 100, and 200 μg/mL CFP and fucoidan for 24 h. The expression of co-stimulators in human MDDCs including CD80 and CD83 was upregulated by the treatment of both CFP and fucoidan in a dose-dependent manner. The expression levels of CD80 and CD83 were considerably higher in the CFP treatment compared with fucoidan (Figure 2A). Moreover, treatment with 100 μg/mL CFP induced substantial increases in the expression of the co-stimulators and MHC class I and II in human MDDCs, which were much higher than those induced by fucoidan (Figure 2B). Furthermore, CFP treatment substantially increased the concentration of cytokines in the cultured medium, with levels nearly comparable to those with the fucoidan treatment (Figure 2C). Taken together, these results suggest that CFP can induce the activation of human MDDCs.

### 2.3. CFP Activates Human PBDCs

We next evaluated the effect of CFP in the activation of human PBDCs. The peripheral blood mononuclear cells (PBMCs) were isolated from the blood of healthy donors and were treated with 100 μg/mL CFP and fucoidan for 6 h. PBDCs were defined as lineage^−^CD11c^+^ cells among live leukocytes and then divided into BDCA1^+^ and BDCA3^+^ PBDCs using flow cytometry (Figure 3A). The CFP treatment induced a substantial increase in the co-stimulators and MHC class I and II expression in both BDCA1^+^ and BDCA3^+^ subsets of PBDC (Figure 3B). In addition, CFP promoted upregulation of pro-inflammatory cytokine production in PBMC-cultured medium (Figure 3C). Consistent with the result of the MDDC experiment, the CFP-induced activation of PBDC subsets was higher than that induced by fucoidan (Figure 3B); however, the production level of proinflammatory cytokines in the CFP treatment did not differ from that in the fucoidan group (Figure 3C). The results suggest that CFP can activate human PBDC subsets.

### 2.4. CFP Activates Human PBDCs

Having established that CFP can activate PBDCs among PBMCs, we next examined whether the CFP directly stimulate PBDCs or whether immune cells contribute to the activation of the PBDCs in PBMCs. BDCA1^+^ and BDCA3^+^ PBDCs were purified and cultured with CFP and fucoidan. The expression levels of co-stimulators and MHC class I and II in both BDCA1^+^ and BDCA3^+^ PBDCs were significantly increased by CFP (Figure 4A,B). Moreover, CFP treatment upregulated the production of IL-6, IL-12, and tumor necrosis factor (TNF)-α in cultured medium of BDCA1^+^ and BDCA3^+^ PBDCs (Figure 4C). Therefore, these data are further evidence that CFP can directly promote the activation of human PBDC subsets.

### 2.5. CFP-Induced Activation of Human PBDCs Required Phosphorylation of p38

Because the mitogen-activated protein kinase signaling pathway activated during maturation of human DCs by stimuli, we next examined whether the activation of this signaling pathway is required for CFP-induced human DC activation. The isolated BDCA1^+^ PBDCs and BDCA3^+^ PBDCs were pre-treated with extracellular signal-regulated kinase (ERK) inhibitor (PD98059), p38 inhibitor (SB203580) and c-Jun N-terminal kinases (JNK) inhibitor (SP600125). One hour after treatment the cells were stimulated by CFP. The expression levels of CD80, CD83 and CD86 were completely abolished by SB203580 in both BDCA1^+^ and BDCA3^+^ PBDCs, whereas PD98059 and SP203580 did not inhibit the CFP-induced upregulation of CD80, CD83 and CD86 (Figure 5A,B). Moreover, CFP promoted IL-12 production in BDCA1^+^ PBDCs and BDCA3^+^ PBDCs were also completely inhibited by pretreatment of SB203580 (Figure 5C,D). Thus, these data suggested that CFP-induced activation of human PBDCs is dependent on activation of p38. 

### 2.6. CFP-Stimulated PBDC Subset Promotes T Cell Immune Activation

The data indicating that that CFP induces activation of PBDC subsets prompted us to examine whether CFP-activated PBDC subsets elicit T cell proliferation and activation. Purified BDCA1^+^ PBDCs were treated with CFP and fucoidan for 6 h and then co-cultured with carboxyfluorescein succinimidyl ester (CFSE)-labeled syngeneic CD4^+^ T cells for another 3 days. The CD4^+^ T cells substantially increased proliferation after co-culturing with CFP-treated BDCA1^+^ PBDCs (Figure 6A,B left panel). The level of interferon-gamma (IFN-γ) production was also greatly increased in CD4^+^ T cells compared to the control group (Figure 6A,B right panel). In addition, CFP-stimulated BDCA3^+^ PBDCs promoted proliferation of CD8^+^ T cells, which increased intracellular production levels of IFN-γ (Figure 6C,D). Moreover, the concentrations of TNF-α and IFN-γ in the cultured medium of CD4^+^ T cells were remarkably elevated by co-culture with CFP-stimulated BDCA1^+^ PBDCs (Figure 6E), which indicates that PBDC promoted development of helper T 1 (Th1) cells. The levels of TNF-α, IFN-γ, and cytotoxic mediator perforin and granzyme B levels in CD8^+^ T cell cultured medium was significantly increased by CFP-activated BDCA3^+^ PBDCs (Figure 6F), which indicates that PBDCs induce differentiation of cytotoxic T lymphocytes (CTLs). These results suggest that CFP-stimulated PBDC subsets elicit Th1 and CTL immune activation. 

## 3. Discussion

Natural products, such as marine polysaccharides, have shown various biological effects in animal models and humans, with comparably less cytotoxicity than synthetic molecular products. Marine polysaccharides, extracted from different species of algae, exhibit similar effects in immune modulation. Fucoidan and ascophyllan extracted from brown algae have been shown to function as adjuvants to enhance immune activation and promote antigen-specific anti-cancer immunity in a mouse model [29,30]. Ulvan, which is the most well-studied polysaccharide derived from green algae, showed similar effects in immune cell activation as those shown by a polysaccharide derived from a brown alga [31,32]. In the present study, we demonstrated an additional immunostimulatory polysaccharide—extracted from a green alga—that induces the activation of human PBDCs. In addition, the CFP-induced PBDC activation promoted Th1 and CTL immune responses against syngeneic T cells. 

Different compositions of monosaccharides in the polysaccharide may elicit different biological activities. Fucoidan from different algae showed comparable effects in immune activation, possibly because the higher composition of galactose may affect the activation of immune cells [2,6,23]. Ascophyllan, which has also shown a stronger effect than fucoidan in murine DC activation, has a higher composition of galactose compared with fucoidan [30]. The monosaccharide composition of CFP is much different from that of fucoidan. Fucoidan contains fucose as the main component, whereas mannose is the main component of CFP. We also found that the effect of CFP in the induction of human DC activation was much stronger than that of fucoidan. That stronger effect of CFP in human DC activation may be due to the different composition of monosaccharides.

The immunomodulatory effect of polysaccharides on immune cells depends on different receptors and signaling pathways. Previous studies have shown that fucoidan activates human PBDCs via scavenger receptor-A (SR-A), which is regulated by the mitogen-activated protein kinase pathway, phosphoinositide 3-kinase (PI3K), glycogen synthase kinase 3, and nuclear factor-κB [11]. Berria et al. found that ulvan induced cytokine production in intestinal epithelial cells via the toll-like receptor 4-mediated PI3K/protein kinase B (Akt) signaling pathway [33]. Previous studies have reported that CFP stimulates NK cells via the complement receptor 3 (CR3) in mouse cells [13]. In murine DC activation, there is still debate concerning whether the ligation of CR3 can induce activation or suppression of DCs [34]. In the case of human DCs, the stimulation of CR3 has been shown to induce the activation of DCs [35,36]. Therefore, CFP may activate human PBDCs through CR3 binding. However, the receptor of CFP and its activating signaling pathway in human DCs are still unclear. It may be that CFP was stronger in inducing the co-stimulatory expression in PBDCs compared to fucoidan, whereas CFP-induced proinflammatory cytokine production was not different to that induced by fucoidan. We plan to further examine what types of receptors and signaling pathways contribute to PBDC activation by CFP. 

Allogeneic MHC molecules promote activation of T cells [12]; therefore, syngeneic T cells must be used in a human DC study to evaluate the maturation and activation of DCs. We evaluated PBDC activation by CFP by examining the proliferation of syngeneic T cells and the production of cytokines. Helper T cells contribute to the production of cytokines, which induces different immune cell activation, such as CTLs, macrophages and NK cells [37]. CTLs produce proinflammatory cytokines and cytotoxic mediators to kill target cells [38]. Since the PBDC subsets are specialized for the activation of different T cells, we found that CFP-stimulated BDCA1^+^ and BDCA3^+^ PBDCs promote syngeneic Th1 and CTL immune responses, respectively. These results provide additional evidence that CFP will be a strong candidate molecule for enhanced DC-mediated T cell immunity in humans.

## 4. Materials and Methods 

### 4.1. Ethics Statement 

This study was conducted in accordance with the Declaration of Helsinki and approved by the Institutional Review Board at Shanghai Public Health Clinical Center (IRB number: 2017-Y037). Elutriated peripheral blood mononuclear cells (PBMCs) were obtained from healthy donors at the Shanghai Public Health Clinical Center. Written informed consent was obtained from all volunteers.

### 4.2. Extraction of CFP

CFP was prepared as previously described [16]. Briefly, a dried and milled *C. fragile* sample was treated with 90% ethanol at room temperature overnight. After the removal of ethanol, the sample was extracted with distilled water at 65 °C for 2 h. The water-soluble crude sample was precipitated with ethanol and filtrated. This was re-suspended in distilled water, and free proteins were removed using the Sevag method. The solution was fractionated by an ion-exchange chromatography with a DEAE Sepharose fast flow column (17-0709-01, GE Healthcare Bio-Science AB, Uppsala, Sweden). Chromatographic separation resulted in three fractions (F1, F2, and F3). The most immunostimulatory polysaccharide, fraction F2, was chosen for further study and designated as CFP [16]. The endotoxin levels in CFP were measured using a Limulus amebocyte lysate (LAL) kit (Lonza, Basel, Switzerland). Fucoidan was obtained from Sigma-Aldrich (St Louis, MO, USA).

### 4.3. Reagents and Antibodies

Fucodian from *Fucus vesiculosus*, PD98059, SB230580 and SP600125 were purchased from Sigma-Aldrich. Anti-CD3 (HIT3a), anti-CD14 (63D3), anti-CD16 (3G8), anti-CD19 (HIB19), anti-CD20 (2H7), anti-CD34 (561), and anti-CD56 (5.1H11), anti-CD80, anti-CD83, anti-CD86, MHC class I and MHC class II were procured from BioLegend (San Diego, CA, USA).

### 4.4. MDDC Generation

Monocytes were isolated from PBMCs using a monocyte isolation kit (Miltenyi Biotec, Bergisch Gladbach, Germany) and cultured for 6 days in Roswell Park Memorial Institute-1640 cultured medium (100 U/mL penicillin/streptomycin), 50 ng/mL rhGM-CSF, and 50 ng/mL rhIL-4 in 10% autologous serum. The upregulation of CD1a and downregulation of CD14 were used as markers for evaluating the differentiation efficiency of MDDCs. Only those cells that were greater than 95% CD1a-positive were considered MDDCs and were used for further experiments (Appendix A). 

### 4.5. Flow Cytometry Analysis

Cells were pre-incubated with fragment crystallizable receptor-blocking antibodies (Abs) and isotype controls for prevention of non-specific binding. The cells were stained with fluorescence-conjugated monoclonal Abs at 4 °C for 20 min. After the unbound Abs were washed, the cells were stained with 4′,6-diamidino-2-phenylindole (DAPI; Sigma-Aldrich). The DC population and its activation marker expression were analyzed using flow cytometry (Becton Dickinson, Franklin Lakes, NJ, USA). 

### 4.6. Analysis and Isolation of PBDCs

PBMCs were stained with lineage marker containing anti-CD3 (HIT3a), anti-CD14 (63D3), anti-CD16 (3G8), anti-CD19 (HIB19), anti-CD20 (2H7), anti-CD34 (561), and anti-CD56 (5.1H11). The PBDCs were defined as CD11c^+^Lineage^−^ cells by fluorescence-activated cell sorting (FACS) Fortessa (Becton Dickinson). BDCA1^+^ PBDCs were isolated from PBMCs by CD1c^+^ (BDCA1^+^) DC Isolation kit (Miltenyi Biotec). BDCA3^+^ PBDCs were isolated by staining with BDCA3 microbead kit (Miltenyi Biotec). The purity of BDCA1^+^ and BDCA3^+^ PBDC preparations was greater than 90%. 

### 4.7. Syngeneic T Cell Proliferation and Activation 

Isolated BDCA1^+^ and BDCA3^+^ PBDCs were stimulated with 100 μg/mL CFP for 6 h. Syngeneic CD4 and CD8 T cells were isolated from PBMCs (Miltenyi Biotec) of the same donor and labeled with 10 μM carboxyfluorescein succinimidyl ester (CFSE) (Invitrogen, San Diego, CA, USA). The BDCA1^+^ PBDCs were co-cultured with CD4 T cells and BDCA3^+^ PBDCs were incubated with CD8 T cells a ratio of 1:25, respectively. Three days after co-culture, the cells were further stimulated for 4 h with 50 ng/mL phorbol 12-myristate 13-acetate (PMA; Merck, Kenilworth, NJ, USA) and 1 μM ionomycin (Merck), with the addition of monensin solution (BioLegend) during the final 2 h. The CFSE dilution and intracellular IFN-γ production in CD4^+^ and CD8^+^ T cells were analyzed using flow cytometry (Becton Dickinson).

### 4.8. Enzyme-Linked Immunosorbent Assay (ELISA)

Concentrations of cytokines in cultured medium were measured in triplicate using ELISA kits. IL-6, IL-12p70, and TNF-α ELISA kits were obtained from BioLegend. Perforin and granzyme B ELISA Kits were purchased from Abcam (Cambridge, UK). 

### 4.9. Statistical Analysis

Data are expressed as mean ± standard error of the mean (SEM). A one- or two-way ANOVA (Tukey multiple comparison test) and the Mann–Whitney t-test were used for analysis of the data sets. *p* values < 0.05 were considered statistically significant. 

## 5. Conclusions

The results of our study confirm our hypothesis that the green alga-extracted polysaccharide, CFP, is a powerful immune activator in human DCs. Moreover, CFP can directly promote the activation of human PBDC subsets, which consequently elicits Th1 and CTL activation. Therefore, CFP may act as a potential immunostimulator and adjuvant in humans.

## Figures and Tables

**Figure 1 marinedrugs-18-00535-f001:**
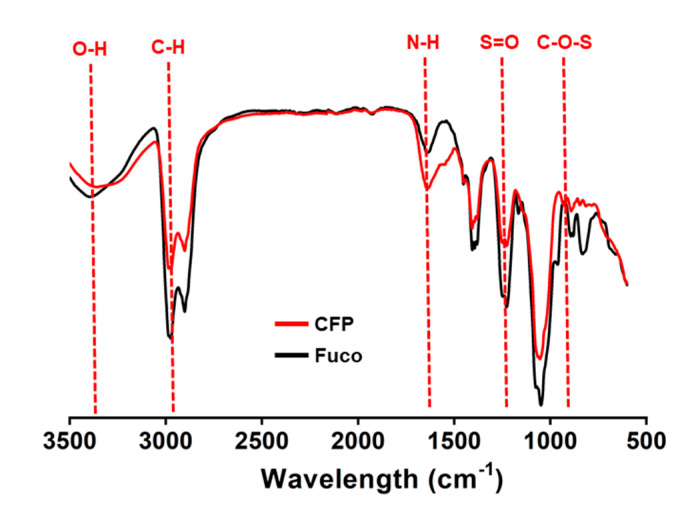
FT-IR spectrum pattern of *Codium fragile* polysaccharide (CFP). Functional groups related with sulfate groups as well as the hydroxyl and the amide bond in CFP were determined by Fourier transform spectroscopy (FT-IR; Spectrum 100, Perkin Elmer).

**Figure 2 marinedrugs-18-00535-f002:**
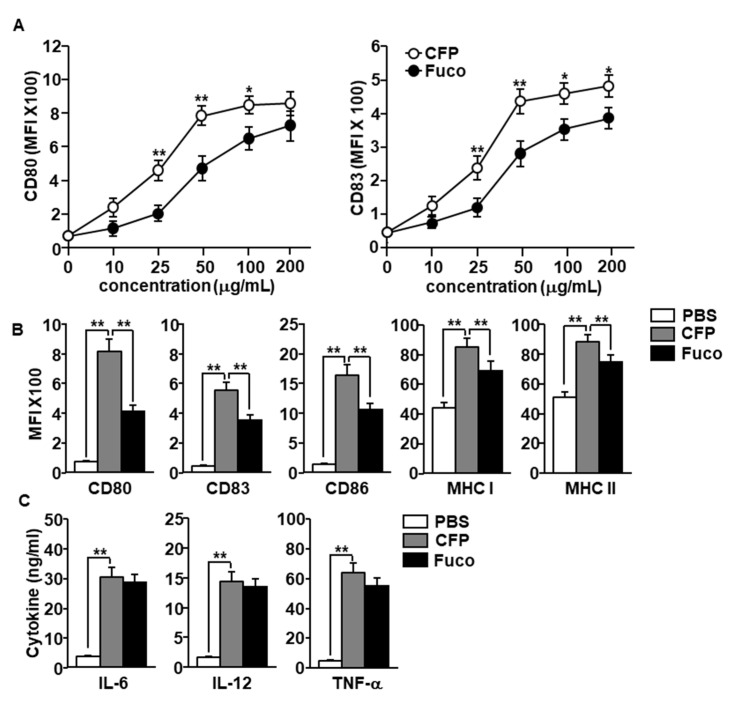
Activation of monocyte-derived dendritic cells (MDDCs) by *C. fragile* polysaccharide (CFP). MDDCs were differentiated from human peripheral blood monocytes by culturing with recombinant human interleukin-4 (rhIL-4) and recombinant human granulocyte-macrophage colony-stimulating factor (rhGM-CSF) for 6 days. (**A**) Dose dependent expression levels of CD80 and CD83 by CFP and fucoidan. (**B**) Mean fluorescence of intensity of co-stimulators and major histocompatibility complex (MHC) class I and II in MDDCs was measured by flow cytometry. (**C**) Concentration of interleukin (IL)-6, IL-12, and tumor necrosis factor (TNF)-α in cultured medium. The data are presented as averages of six independent blood samples for each group. * *p* < 0.05, ** *p* < 0.01.

**Figure 3 marinedrugs-18-00535-f003:**
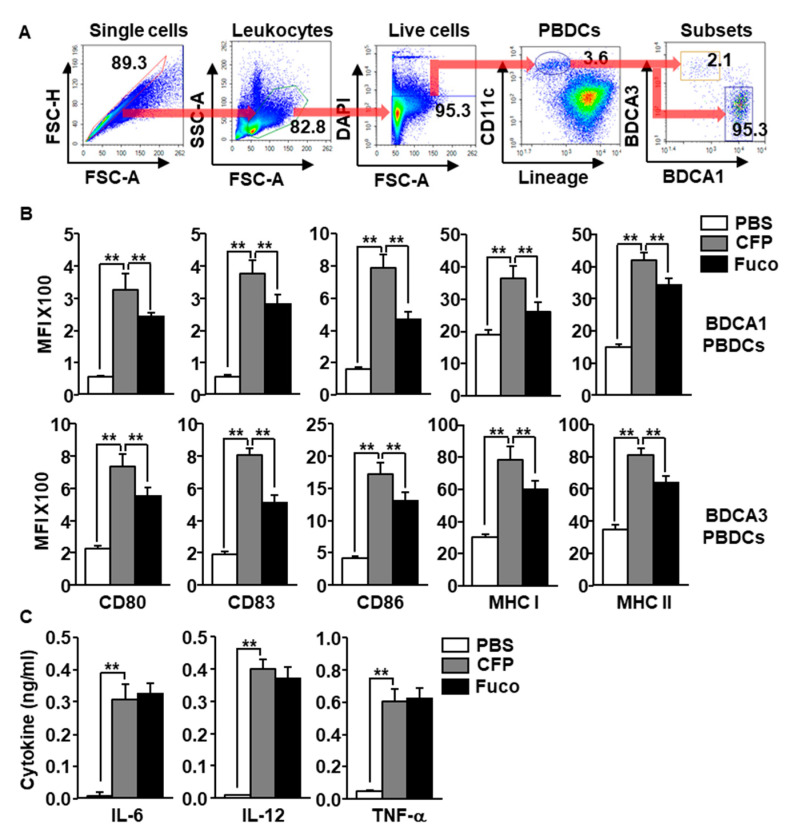
CFP promoted activation of human peripheral blood dendritic cells (PBDCs). Peripheral blood mononuclear cells (PBMCs) were treated with 100 μg/mL CFP and 100 μg/mL fucoidan. Six hours after treatment, the cells were harvested and the activation of blood dendritic cell antigen 1^+^ (BDCA1^+)^ and BDCA3^+^ PBDCs was assessed. (**A**) Subsets of human PBDCs were defined as BDCA1^+^ and BDCA3^+^ cells in lineage^−^CD11c^+^ live leukocytes. (**B**) Expression levels of co-stimulators and MHC class I and II in BDCA1^+^ PBDCs (upper panel) and BDCA3^+^ PBDCs (lower panel). (**C**) Production levels of IL-6, IL-12, and TNF-α were measured in culture medium. All data are presented as averages and representative of six independent blood samples. ** *p* < 0.01.

**Figure 4 marinedrugs-18-00535-f004:**
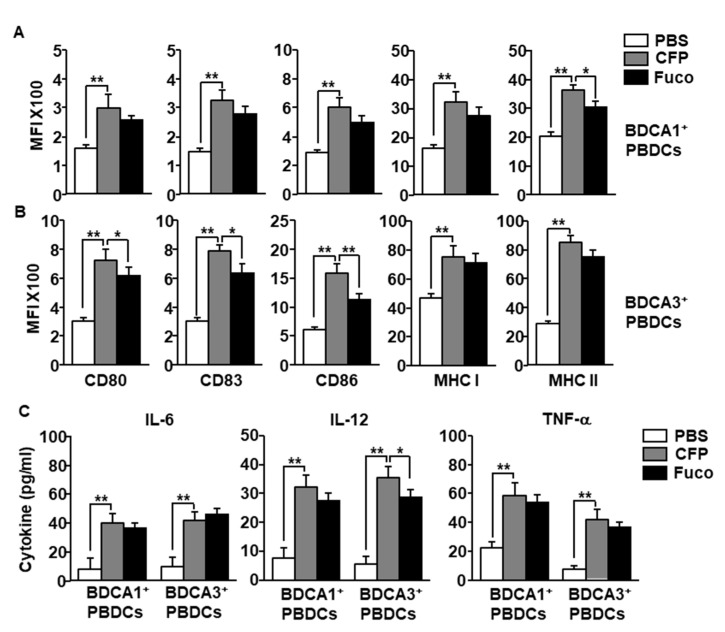
CFP promotes activation of isolated PBDC subsets. (**A**) BDCA1^+^ PBDCs were isolated from PBMCs and treated with 100 μg/mL CFP and fucoidan. The expression levels of co-stimulators and MHC class I and II were measured 6 h after treatment. (**B**) Isolated BDCA3^+^ PBDCs were treated with 100 μg/mL CFP, and the expression of co-stimulators and MHC class I and II expression were measured 6 h after treatment. (**C**) The concentrations of IL-6, IL-12, and TNF-α in cultured medium were measured by enzyme-linked immunosorbent assay (ELISA). Data are averages from six independent blood samples for each group. * *p* < 0.05, ** *p* < 0.01.

**Figure 5 marinedrugs-18-00535-f005:**
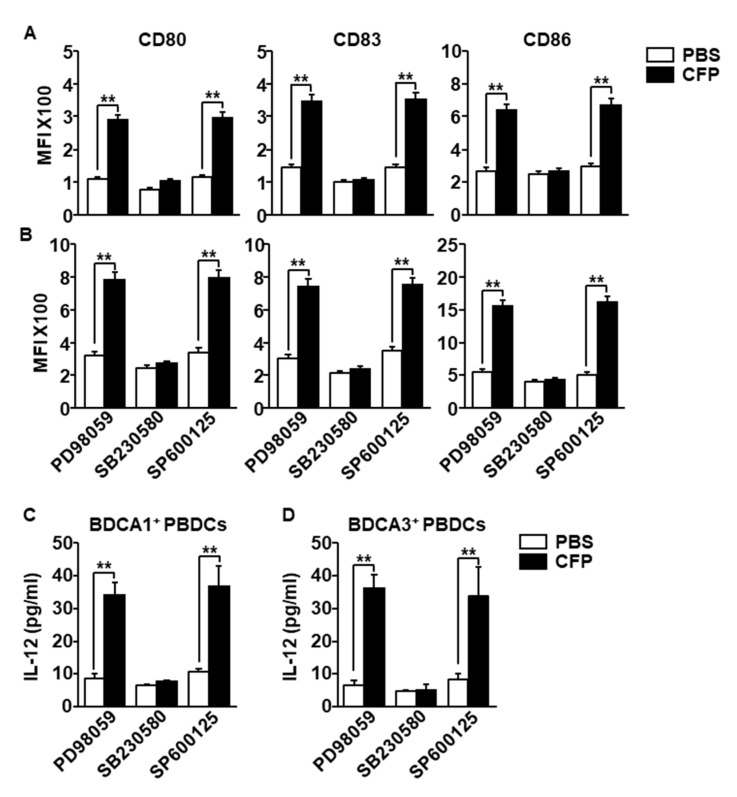
Activation of p38 was required for CFP-induced human PBDC activation. The isolated BDCA1^+^ PBDCs and BDCA3^+^ PBDCs were pretreated with PD98059 (10 μM), SB203580 (50 μM) and SP600125 (10 μM). One hour after treatment, the cells were incubated with 100 μg/mL CFP for 24 h. (**A**) CD80, CD83 and CD86 expression levels in BDCA1^+^ PBDCs were measured by a flow cytometer. (**B**) CD80, CD83 and CD86 expression levels in BDCA3^+^ PBDCs were shown. (**C**,**D**) The concentrations of IL-12 in (**C**) BDCA1^+^ PBDC and (**D**) BDCA3^+^ PBDCs cultured medium were measured by ELISA. Data are averages from six independent blood samples for each group. ** *p* < 0.01.

**Figure 6 marinedrugs-18-00535-f006:**
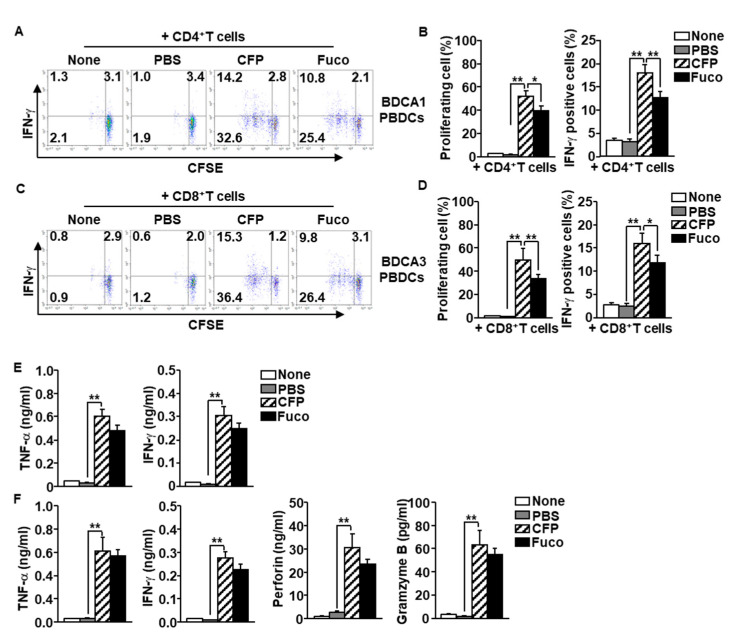
Activation of T cells by CFP-stimulated PBDC subsets. BDCA1^+^ and BDCA3^+^ subsets of PBDCs were treated with PBS, CFP, and fucoidan, respectively. Six hours after stimulation, the PBDCs were co-cultured with syngeneic CD4^+^ or CD8^+^ T cells, respectively. (**A**) Carboxyfluorescein succinimidyl ester (CFSE) dilution and intracellular interferon (IFN)-γ production in CD4 T cells by co-culture with BDCA1^+^ PBDCs were analyzed by flow a cytometer. (**B**) Mean percentage of proliferating CD4^+^ T cells (left panel). Mean positive cells of IFN-γ-producing cells (right panel). (**C**) Proliferating and IFN-γ-producing CD8^+^ T cells by co-culture with BDCA3^+^ PBDCs were measured by flow cytometry (left panel). (**D**) Mean percentage of proliferating CD8^+^ T cells (left panel). Mean positive cells of IFN-γ-producing cells (right panel). (**E**) Concentration of tumor necrosis factor (TNF)-α and IFN-γ producing levels in CD4^+^ T cells and BDCA1^+^ PBDCs co-cultured medium. (**F**) Production levels of TNF-α, IFN-γ, perforin, and gramzyme B were measured in the co-cultured medium of CD8^+^ T cells and BDCA3^+^ PBDCs. Data are representatives and averages of four independent blood samples for each group. * *p* < 0.05, ** *p* < 0.01.

**Table 1 marinedrugs-18-00535-t001:** Compositions (*w/w*%) of neutral sugar.

Polysaccharide Source	Composition of Neutral Sugar ^a^					Uronic Acid ^b^	SO_4_^2− c^
	Fucose	Xylose	Glucose	Mannose	Galactose		
***Coidum fragile***	-	-	5.79	61.5	-	2.4	10.3
***Fucus vesiculosus***	38.02	2.73	0.49	1.27	3.38	5.49	24.53

^a^ Determined by HPLC after acidic hydrolysis [24,25]. ^b^ Determined by carbazole method and calculated as glucuronic acid equivalent [24,26]. ^c^ Determined by turbidimetric assay after acidic hydrolysis [27,28].

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
