# Peer review of "Human Peripheral Blood Dendritic Cell and T Cell Activation by Codium fragile Polysaccharide"

_marinedrugs, 2020, doi:10.3390/md18110535_

Round 1

Reviewer 1 Report

General comments:

1.      How to confirm the immunostimulatory polysaccharide, fraction 2?

2.      Please present the various functional groups in the Fig. 1.

3.      Recheck the style of writing and the abbreviation.

4.      No prove the cell characteristics of MDDCs.

5.      The activating signaling pathway in human DCs by CFP should be proved.

Author Response

Reveiwer 1

General comments:

  1. How to confirm the immunostimulatory polysaccharide, fraction 2?

Answer (A): Thank you for the reviewer’s comments. The immune stimulatory effect of different faction of polysaccharide from C. fragile has evaluated previous study as shown in reference #16.

  1. Please present the various functional groups in the Fig. 1.

A: We marked the functional groups related with sulfate groups as well as the hydroxyl and the amide bond in Fig.1

  1. Recheck the style of writing and the abbreviation.

A: We have no carefully check writing style and the abbreviation. Moreover, we added the abbreviation in the end of the mansucript.

  1. No prove the cell characteristics of MDDCs.

A: Differentiation of MDDCs were determined by expression levels of CD1a. We have now added the expression levels of CD1a of MDDCs in the supplemental figure 2. 

  1. The activating signaling pathway in human DCs by CFP should be proved.

A: We thanks for the important comments. Since MAP Kinase pathway is the main signaling pathway for DC activation, we examined the effect of MAP Kinase inhibitor in the CFP-induced activation of PBDCs. As shown in Figure 5, p38 inhibitor abolished CFP-induced activation of human DCs, which indicates the CFP-induced activation of human DCs was dependent on p38 activation.  

Reviewer 2 Report

Human peripheral blood dendritic cell and T cell activation by Codium fragile polysaccharide

Wei Zhang1#, Juyoung Hwang1,2,3#, Hae-Bin Park1,2,3, Seong-Min Lim2,3, Seulgi Go2,3, Jihoe Kim2,3, 4 Inho Choi2,3, SangGuan You4*, and Jun-O Jin1,2,3*

Synopsis: In this manuscript by You, Jin et al., the authors demonstrate that the sulfated polysaccharide from C. fragile (CFP) possesses the ability to upregulate various molecular markers and cytokines responsible for the activation of dendritic cells, followed by both helper and cytotoxic T cells. The activity of CFP was compared against a fucoidan isolated from F. vesiculosus and was found to be statistically superior for many of the biological markers studied. This difference in activity was hypothesized to be due to the differenced in monosaccharide composition, where CFP is enriched in Man, while the fucoidan had high levels of Fuc, and to a lesser extent, Gal. The authors postulate that CFP may be beneficial as an adjuvant in the activation of immune function of humans.

This is a well-written paper that shows statistically significant results for most of the biomarkers analyzed for the various bioassays conducted, however, it could be a stronger manuscript with a few updates as suggested below, most of which relate to the presentation/discussion of the material in the figures and tables.

Major issues:

  • Is there cytotoxicity data for CFP? If so, it should be included in this paper. Also, if there is cytotoxicity data for the fucoidan, it should be presented side-by-side with CFP.
  • In Table 1, subscripts a-c, these methods are neither mentioned in the experimental methods, nor are any references provided for the published methods where needed (b,c).
  • For Figures 2-5, the following need to be addressed.
    • The captions for the bar graphs all need to state what is being studied (with definitions for new abbreviations used) relative to PBS control.
    • Also for the bar graphs, some captions are missing definitions of statistical significance at the end (*p <05, **p <0.01.) Figures 4 and 5.
    • Bar graphs: For the statistical significance on top of each bar, please indicate in the caption what it signifies, as this is unclear. Is it compared to the PBS control, or is it a comparison between CFP and the fucoidan? If both are shown, there needs to be a clear distinction between the two measures. It is vague and confusing as it is represented currently.
    • For the flow cytometry results, please indicate what the difference is in the sections identified in the area of the plots. It is not enough to say that different cell types were identified (Figure 3). For figure 5, A and B, a left, middle and right panel are mentioned but there are 4 panels, so this is vague. Please clarify what is being identified for each.

Minor Issues:

  • There are many abbreviations that are not defined anywhere in the text. I think it would be very helpful to the readers to include a list of these abbreviations at the end so that readers can remind themselves what the abbreviations are. For example: BDCA1+ and BDCA3+, rhGM-CSF, CFSE-l, PI3K/Akt, FSCs, DAPI, MFIX100, etc.. All abbreviations should also be defined on first use in the text. Some, like rhGM-CSF are defined after the first use.
  • In Figure 1, please label the signals associated with the sulfate signals in the IR.
  • Minor English usage:
    • Line 61: MDDCs are a single type of cells should be replaced with, “MDDCs represent a single cell type,”

Author Response

Reviewer 2

Synopsis: In this manuscript by You, Jin et al., the authors demonstrate that the sulfated polysaccharide from C. fragile (CFP) possesses the ability to upregulate various molecular markers and cytokines responsible for the activation of dendritic cells, followed by both helper and cytotoxic T cells. The activity of CFP was compared against a fucoidan isolated from F. vesiculosus and was found to be statistically superior for many of the biological markers studied. This difference in activity was hypothesized to be due to the differenced in monosaccharide composition, where CFP is enriched in Man, while the fucoidan had high levels of Fuc, and to a lesser extent, Gal. The authors postulate that CFP may be beneficial as an adjuvant in the activation of immune function of humans.

 This is a well-written paper that shows statistically significant results for most of the biomarkers analyzed for the various bioassays conducted, however, it could be a stronger manuscript with a few updates as suggested below, most of which relate to the presentation/discussion of the material in the figures and tables.

 Answer (A): Thank you for the crucial comments.

Major issues:

  • Is there cytotoxicity data for CFP? If so, it should be included in this paper. Also, if there is cytotoxicity data for the fucoidan, it should be presented side-by-side with CFP.

A: the cytotoxicity of CFP and fucoidan was examined in the RAW264.7 and HL-60 cells and found it did not promoted any cytotoxicity in those cells (Figure S1).

  • In Table 1, subscripts a-c, these methods are neither mentioned in the experimental methods, nor are any references provided for the published methods where needed (b,c).

A: We have now added references in Table 1.

  • For Figures 2-5, the following need to be addressed.
    • The captions for the bar graphs all need to state what is being studied (with definitions for new abbreviations used) relative to PBS control.

A: We have now revised the figure legend.

  • Also for the bar graphs, some captions are missing definitions of statistical significance at the end (*p <05, **p <01.) Figures 4 and 5.

A: We sorry for the error. We have revised the figure legend.

  • Bar graphs: For the statistical significance on top of each bar, please indicate in the caption what it signifies, as this is unclear. Is it compared to the PBS control, or is it a comparison between CFP and the fucoidan? If both are shown, there needs to be a clear distinction between the two measures. It is vague and confusing as it is represented currently.

A: We revised the Figures for a clear distinction between the experimental groups.

  • For the flow cytometry results, please indicate what the difference is in the sections identified in the area of the plots. It is not enough to say that different cell types were identified (Figure 3). For figure 5, A and B, a left, middle and right panel are mentioned but there are 4 panels, so this is vague. Please clarify what is being identified for each.

A: Sorry to make confusing. We have added arrows in the dot plots for indicating analysis strategy of the PBDC subsets in Figure 3. In Figure 5, we have now separated the figure as A and B.

Minor Issues:

  • There are many abbreviations that are not defined anywhere in the text. I think it would be very helpful to the readers to include a list of these abbreviations at the end so that readers can remind themselves what the abbreviations are. For example: BDCA1+ and BDCA3+, rhGM-CSF, CFSE-l, PI3K/Akt, FSCs, DAPI, MFIX100, etc.. All abbreviations should also be defined on first use in the text. Some, like rhGM-CSF are defined after the first use.

A: Thank you for the comments. We revised the manuscript and also added abbreviation in the end of the text.

  • In Figure 1, please label the signals associated with the sulfate signals in the IR.

A: As reviewer suggested, we labeled the signals related with the sulfate groups in Fig. 1

  • Minor English usage:

A: We have received English editing service from Editige.

  • Line 61: MDDCs are a single type of cells should be replaced with, “MDDCs represent a single cell type,”

A: We have now changed the sentence.